# Bioaccumulation Levels and Potential Health Risks of Mercury, Cadmium, and Lead in Albacore (*Thunnus alalunga*, Bonnaterre, 1788) from The Aegean Sea, Greece

**DOI:** 10.3390/ijerph16050821

**Published:** 2019-03-06

**Authors:** Nikolaos Stamatis, Nikolaos Kamidis, Pelagia Pigada, Despoina Stergiou, Argyris Kallianiotis

**Affiliations:** Hellenic Agricultural Organisation—Demeter, Fisheries Research Institute, N. Peramos, 64007 Kavala, Greece; nikkami@inale.gr (N.K.); pigada@inale.gr (P.P.); stergiou@inale.gr (D.S.); homekall@otenet.gr (A.K.)

**Keywords:** fish, albacore tuna, toxic metals, THQs values, health risks assessment, Greece

## Abstract

Aegean Sea albacore (*T. alalunga*), fresh or processed, is marketed locally in Greece or exported, mainly to Japan, Italy, Spain, and France. To provide information for consumers and biomonitoring programs and assess the potential human health risks, concentrations of mercury (Hg), cadmium (Cd) and lead (Pb) were determined in albacore edible muscle samples from two fishing grounds of the Aegean Sea, Greece via graphite furnace atomic absorption spectrometry (GF-AAS). Of the 82 individuals, 28 contained Cd and three contained Pb above the permissible limits set by the European Union (0.1 mg kg^−1^ wet wt and 0.3 mg kg^−1^ wet wt, respectively). None of the samples contained mercury above the limit (1.0 mg kg^−1^ wet wt). Potential health risks to human via dietary intake of albacore were estimated by the total target hazard quotients (TTHQs), which indicated that the consumers could acquire health problems due to consumption of Aegean Sea albacore. Thus consequently, concentrations of toxic heavy metals in albacore, especially mercury, must be monitored regularly and comprehensively with respect to consumer health.

## 1. Introduction

Albacore (*T. alalunga*) is a pelagic large predator of high ecological and commercial interest. It comprised one of the economic tuna species, particularly popular in the fish exploitation (canning) sector and the sushi market [1]. Six independent stocks are supposed to be distributed in the Mediterranean Sea and the three major oceans, between latitudes 50° N and 40° S. Current catches correspond to 4% of the global tuna fisheries [2]. The global tuna catch has begun to stagnate, and the stocks of the most popular species such as *T. alalunga* are now completely or heavily exploited [3]. The Mediterranean albacore stock is managed by the International Commission for the Conservation of Atlantic Tunas (ICCAT) that conducted assessments of this species in 2017, which were performed using methods for data-poor stocks [4]. Currently, the status of the Mediterranean albacore remains highly uncertain and it is being exploited without any conservation measures, e.g., limits on total allowable catches, capacity limitations, sport and recreational fisheries, technical characteristics of the fishing gears or minimum lengths [5]. In Greece, fishing of albacore takes place mainly in the northern part of the Aegean Sea from August until November with troll lines and long lines [6]. While the amount of Greek catch in 2006 was 401 tons, it reached 1332 tons in 2016 [4], and albacore became a valuable export product for Greece.

Tuna fish, like other marine fish species, is very important in the human diet and represents the main source of protein, polyunsaturated fatty acids (PUFAs), vitamins and a wide range of other important nutrients [7,8,9,10,11]. However, large predators, such as albacore tuna, are at the top of the aquatic food chain, and hence they can bioaccumulate large amounts of toxicants, such as the heavy metals mercury (Hg), cadmium (Cd) and lead (Pb) [12,13,14,15]. In addition, albacore tuna is a high-performance fish with very high metabolitic rates, which contributes to the enhanced bioaccumulation of such metals [16]. Mercury, Cd, and Pb as non-essential elements can be very harmful even at low concentrations, occupying top positions in the lists of toxicants and recognized for their toxicity towards most organisms [17,18], particularly fish and other seafood [19,20]. Their effects on human health were always of great interest. Due to their toxicity, Hg, Cd, and Pb have been the subject of major studies, and their presence in marine food must be carefully checked to prevent possible toxicological risks [21]. Mercury is a toxic element to human and fish are acknowledged to be the largest source of Hg for humans. Individuals eating high quantities of tuna may have increased health risks such as developmental anomaly of the brain of infants and neurological problems in adults [22]. Cadmium may induce skeletal weakness, reproductive deficiencies, neurological and digestive disorders, cancers, mutations, and kidney dysfunction [7,23,24,25]. Lead is another known toxicant for human considered as a toxic element due to its biochemical effects which includes hypertension, hematological effects, neurological problems and renal dysfunction [23]. It is also known to induce reduced cognitive development in children and diseases on cardiovascular, digestive, immune or urinary to multiple organ systems in adults [26,27].

The Aegean Sea (Figure 1) is a semi-enclosed system with ever-growing anthropogenic pressure along its coastline [28,29] that features several highly populated cities with widely urbanized and industrialized areas. Evidences of pollution exist regarding toxic heavy metals, e.g., [30,31,32] and nutrients e.g., [33]. However, publications on the concentrations and the human health risks assessment of Hg, Cd, and Pb in albacore from the Aegean Sea region are very limited [34]. Therefore, the objective of the present work was to study the concentrations of Hg, Cd, and Pb in albacore tuna (*T. alalunga*) samples from two fishing grounds in the Aegean Sea, in order to evaluate the possible risks for human health and to ensure that the fish, fresh or processed, is safe for human consumption. In particular, it was verified whether the concentrations of these contaminants fulfill the criteria set by the European Union (EU) regulation for commercialized food [35] or whether the estimated TTHQ level satisfies the standards established by food safety organisations [36]. Moreover, this paper provides useful information on the impact on the biotic system according to the European Directive 2008/56/EC (Marine Strategy Framework Directive, MSFD), particularly explores descriptor 9 (Contaminants in fish and other seafood for human consumption do not exceed levels established by the Community) of the MSFD directive.

## 2. Materials and Methods

### 2.1. Sampling

In August until November 2015, during several surveys, in total 82 specimens of albacore tuna (weight range: 3.07–12.52 kg, mean: 7.94 ± 2.37 kg; length range: 57.01–101.52 cm, mean: 79.26 ± 9.98 cm) were caught in two fishing grounds of the Aegean Sea by domestic fishermen (Figure 1). From each of the fishing grounds called “North Aegean Sampling Station Area” (NASSA) and “Southeastern Aegean Sampling Station Area” (SASSA) 41 individuals were caught, respectively. Fish landings for the NASSA and SASSA specimens took place at the ports of “Porto Koufo” (Chalkidiki Peninsula) and “Kalymnos Island” (Southeastern Aegean Sea), respectively.

A clean plastic knife was used to remove samples of axial muscle from the left dorsal area above the lateral line and anterior to the origin of the first dorsal fin. This region is representative of the portion of fish consumed by humans [37]. The muscle tissues were immediately placed in sterile plastic polyethylene bags and kept in a deep freeze at −28 °C until chemical analysis.

### 2.2. Heavy Metals Analyses and Quality Control

In the laboratory, subsamples (about 2 g) were homogenized (T-25 Classic Homogenizer, IKA, Staufen im Breisgau, Germany) and digested in a HNO_3_-HClO_4_ mixture in a Q-6000 microwave digestion system (Questron, Mississauga, Canada). The resultant solutions were then diluted to a 25 mL volume in an Erlenmeyer flask with double distilled deionized water. Quantitative determinations of Pb, Cd, and Hg were made using an Analyst 800 atomic absorption spectrometer (Perkin Elmer, Überlingen, Baden-Württemberg, Germany) equipped with a transversely heated graphite furnace (THGA) and longitudinal Zeeman-effect background corrector, AS-800 autosampler, 8-lamp turret, PE AA accessory cooling system and the WinLab32 PC work station. The fish muscles were handled with plastic materials that were washed with 2N HNO_3_ and rinsed with double distilled deionized water, in order to avoid any metal contamination. Acid washed glassware; analytical grade reagents and double distilled deionized water were used in the tissue analysis. The analyses were carried out in triplicate, and the significance level was chosen as 0.05. Detection limits (µg/kg^−1^) were 0.6, 0.002 and 0.05 for Hg, Cd, and Pb respectively. Analytical quality control was achieved using the certified reference material, DORM-2 Dogfish Muscle (National Research Council of Canada). The recovery percentage of Hg, Cd, and Pb concentrations from the reference material was (%) 98.15, 97.61 and 96.32, respectively. In order to determine the precision of the analytical process, samples from two individuals were analyzed four times. The standard deviation for both samples was calculated to (%) 2.5 and 3.1, respectively, and can be considered satisfactory for such kind of environmental analysis. 

### 2.3. Risk Assessment Analyses

The target hazard quotients (THQs) index was applied to assess the potential health risk associated with consumption of the albacore species of marine specimens sampled. The THQ value was calculated on the base of the metal concentrations recorded in the edible fish muscle of the individuals. THQ values >1 indicate a potential health risk to the consumers [38]. It was calculated following the US-EPA [38] formula:(1)$$\mathrm{THQ}=\frac{\mathrm{EF}\times \mathrm{ED}\times \mathrm{FIR}\times C}{\mathrm{RfD}\times \mathrm{WAB}\times \mathrm{TA}}\times 10^{-3}$$
where: EF and ED represent the exposure frequency (365 days/year) and the average lifetime duration (70 years), respectively; FIR is the fish ingestion rate (36 g day^−1^ for person; [39]); C is the metal concentration (mg kg^−1^ wet wt); RfD is the reference oral dose in mg kg_bw_^−1^ d^−1^ (1 ×10^−4^ for Hg, 1 × 10^−3^ for Cd, 4 × 10^−3^ for Pb) [40]; WAB is the average body weight for adult consumer (67 kg); TA is the average exposure time (365 days/year × ED).

Since the exposure to two or more toxic metals may result in additive effects [41], the total target hazard quotient (TTHQ) was also calculated as the arithmetic sum of each THQ values [42,43]:TTHQ = THQ_Hg_ + THQ_Cd_ + THQ_Pb_(2)

### 2.4. Statistics

Descriptive statistics (mean, range, standard deviation) and correlations between elements and sampling areas were performed using the Statistica 10 software (Statsoft Inc., Tulsa, OK Oklahoma, U.S.A.). The results obtained were expressed as the mean ± standard deviation (SD) and were analyzed by means of analysis of variance (ANOVA). Due to the lack of data normality, a Spearman non-parametric test was performed to assess the differences (between all parameters, biological and chemical) in the two independent NASSA and SASSA sample regions. A *p* < 0.05 was considered to indicate statistical significance.

## 3. Results

### 3.1. Descriptive Statistics and Biological Data

Albacore’s biological data and metal contents are shown at the Appendix A, respectively. Summary statistics for metal concentrations in the edible muscle of each species caught from two different fishing grounds of the Aegean Sea (NASSA and SASSA albacores) are presented in Figure 2 by a box and whisker plot. This box-whisker plot shows the mean (small box in the plot) of the metal contents surrounded by a larger box (±1 times the standard deviation).

The “whiskers” in the plot represent a “95% confidence interval” defined as the category mean ±1.96 times the category standard deviation (normal distribution of concentrations of each metal and each sampling area). Mean values of Pb show no significant differences between NASSA and SASSA albacore samples (ANOVA, *p* < 0.05), whereas those of Cd and Hg were significantly higher (*p* < 0.05) in the NASSA compared to SASSA samples.

Length and weight frequency distributions of albacore organisms are presented in Figure 3 and Figure 4, respectively.

According to the Figure 3 and Figure 4, NASSA samples were obtained from organisms with higher length (mean: 87.695 ± 5.208 cm) and weight (mean: 9.895 ± 1.192 kg) values compared to the SASSA samples (mean length and weight by 70.815 ± 5.359 cm and 5.976 ± 1.428 kg, respectively).

According to the Spearman non-parametric test (*p* < 0.05) in the NASSA dataset, the fish length correlated positively to fish weight, and Cd- and Hg-content (Appendix A). The sex while correlated positively to the maturing stage, it was correlated negatively to Hg content. In the SASSA dataset, the fish length correlated positively to weight, sex and Hg content, while weight correlated positively to sex, and Hg content (Appendix A).

### 3.2. Heavy Metals Content in the Marine Organisms

Mean metals content (±SD) measured in the edible muscle of the albacore organisms are reported in Figure 5, Figure 6 and Figure 7 for Hg, Cd, and Pb, respectively.

#### 3.2.1. Mercury

In the present study, all the albacore tuna analysed contained Hg (Figure 5). Mercury levels ranged between 0.249 and 0.938 mg kg^−1^ wet wt (mean: 0.480 ± 0.152 mg kg^−1^ wet wt) in NASSA, significantly higher (ANOVA, *p* < 0.05) to those detected in the SASSA samples (from 0.141 to 0.753 mg kg^−1^ wet wt; mean: 0.401 ± 0.154 mg kg^−1^ wet wt). The legal EU maximum limit of Hg was set at 1.000 mg kg^−1^ wet wt for fish [35]. In the present study, none of the samples contained Hg above EU maximum limit. However, fourteen NASSA and eleven SASSA albacores (in total 30.5%) had concentrations above 0.500 mg kg^−1^ wet wt.

#### 3.2.2. Cadmium

As shown in Figure 6, all Aegean Sea albacore samples contained Cd. Detected cadmium levels ranged from 0.022 to 0.669 mg kg^−1^ wet wt (mean: 0.184 ± 0.176 mg kg^−1^ wet wt) in NASSA, significantly higher (ANOVA, *p* < 0.05) to those measured in the SASSA samples (min: 0.021; max: 0.558; mean: 0.082 ± 0.118 mg kg^−1^ wet wt). Twenty NASSA and eight SASSA samples (in total 34.2%) had concentrations above permitted EU limit for cadmium in fish (0.100 mg kg^−1^ wet wt) [35]. However, the maximum level of Cd in fish permitted by the Food and Agriculture Organization is 0.500 mg kg^−1^ wet wt [44]. Only two NASSA samples and two SASSA samples exceeded the FAO permitted levels for Cd (Figure 6).

#### 3.2.3. Lead

In the present study, 26 (31.7%) out of 82 albacores contained no Pb (Figure 7). Lead concentrations of NASSA samples were detected to vary between 0.021 and 0.557 mg kg^−1^ wet wt (mean: 0.049 ± 0.116 mg kg^−1^ wet wt), not significant different (ANOVA, p < 0.05) from those detected for SASSA albacore samples (from 0.020 to 0.422 mg kg^−1^ wet wt; mean: 0.046 ± 0.077 mg kg^−1^ wet wt). The maximum lead EU level permitted for fish is 0.300 mg kg^−1^ wet wt of tissue [35]. As presented in Figure 7, two NASSA samples with mean values 0.557 and 0.488 mg kg^−1^ wet wt and one SASSA sample with mean values of 0.422 mg kg^−1^ wet wt contained Pb above the limit.

### 3.3. THQs of Mercury, Cadmium and Lead and TTHQs

Mercury THQs values, ranging from 1.338 to 5.040 in NASSA albacores and from 0.758 to 4.046 in SASSA albacores (Table 1) were calculated at high levels (all >1, except two values for SASSA albacores). Moreover, the THQs of Cd from 0.012 to 0.359 and from 0.011 to 0.316 in NASSA and SASSA samples respectively from consumption of fish being <1, suggested that health risk was insignificant. Analogously, as shown in Table 1, there were no THQ values for Pb >1 through the consumption of NASSA and SASSA albacores, indicating health risk was absent. TTHQs were ranged between 1.353 and 5.213 or 0.078 and 4.058 for NASSA or SASSA albacore samples, respectively and indicated that the consumers could acquire health problems due to consumption of Aegean Sea albacore.

## 4. Discussion

Analysis of Hg, Cd, and Pb in the edible muscle tissues of albacore tuna species from Northern and Southeastern Aegean Sea waters in the Mediterranean Sea showed some interesting variations between the 82 specimens and the two fishing areas. Some individuals showed Cd and Pb levels exceeding the permitted EU levels for these elements. With an exception of two values, belonging to SASSA albacores, mercury THQs values were calculated >1, indicating potential health risks for the consumers. Therefore, we discussed the metals bioaccumulation and the possible human health risks among individuals and sampling areas, their compatibility with results reported from previous studies conducted in the Mediterranean Sea or other marine areas of the world and their probable reasons.

### 4.1. Metals Bioaccumulation

Unfortunately, very limited information exists regarding Hg in albacore from Aegean Sea waters. Mercury concentrations in Aegean Sea albacores from the present study, which are detected under the permissible EU limit for Hg, resulted comparablely lower to data previously reported for other Mediterranean albacores (from 0.880 to 2.340 mg kg^−1^ wet wt; mean: 1.560 ± 0.490 mg kg^−1^ wet wt [45,46,47]). The finding of the present study, that the NASSA samples with a higher mean length and weight have significantly higher mean Hg concentrations compared to the SASSA samples, is in agreement with data from previous studies which showed that larger, older fish have generally higher concentrations than smaller, younger fish. For example, Storelli et al. [46] reported that mercury levels and individual size were highly correlated for bluefin tuna (*Thunnus thynnus*) from the Mediterranean Sea, as well as other researchers found positive correlation in yellowfin (*Thunnus albacares*) and skipjack tuna (*Katsuwonus pelamis*) caught in the Western Indian Ocean [48]. A similar correlation was reported by Besada et al. [49] and Agusa et al. [50] in *T. alalunga*, *T. albacares* and *Thunnus obesus* organisms from Atlantic Ocean and South China Sea, respectively. As presented in Figure 5, all samples contained Hg below EU limit (1.000 mg kg^−1^ wet wt). As well known, Mediterranean albacores exhibit the highest Hg contamination among albacore populations (e.g., New Zealand area: from 0.140 to 0.210; mean: 0.170 mg kg^−1^ wet wt [51]; Spanish Atlantic area: from 0.118 to 0.564; mean: 0.190 mg kg^−1^ wet wt [49]; Northeast US Pacific area: from 0.027 to 0.260; mean: 0.140 ± 0.050 mg kg^−1^ wet wt [52]; Western/Central Pacific area: from 0.239 to 1.180; mean: 0.445 ± 0.148 mg kg^−1^ wet wt [53]). Moreover, it was reported that muscle total mercury concentrations are up to tenfold higher in <10 kg Mediterranean albacores than Pacific albacores of a similar size [46,47]. Such high Hg levels found in the Mediterranean albacore may be attributed to the pollution from Hg mining wastewater in the sea [54]. Also, the above of 0.500 mg kg^−1^ wet wt Hg concentrations of the present study found mostly in the NASSA albacore samples, could be derived due to anthropogenic pressure along North Aegean coastline, in which exist coastal mining operations in Stratoni bay as well as several highly populated cities on the coastal zone (e.g., Thessaloniki, Kavala, Alexandroupolis), with widely industrialized and urbanized areas [28,29,30,31,32]. Furthermore, Evros, Nestos, Strymon, Axios, Loudias, and Aliakmonas river discharges, containing both natural and anthropogenic Hg, can contribute to increasing Hg contents of NASSA albacore.

Similar to the results of our study, Cd concentrations of *T. alalunga* samples from the Adriatic Sea (central Mediterranean) may be above the limit set by EU for Cd [47]. Besada et al. [49] studied the heavy metal levels of three different tuna species from the Atlantic Ocean and reported the highest Cd level to be in *T. alalunga*. However, Storelli et al. [55] reported 0.040 mg kg^−1^ wet wt as the maximum level of Cd in *T. thynnus* samples from the Ionian Sea in the Mediterranean Sea. Nevertheless, much higher Cd levels in tunas from Ecuador (2.4 mg kg^−1^ wet wt mean value and a range from <0.011 to 17.000 mg/kg^−1^ wet wt) were reported by Araújo and Cedeño-Macías [56] possibly due to emissions from industrialized countries and the volcanic activity of the Eastern Pacific Ocean. Other studies in samples of diverse fishing areas do not show such high levels of Cd, but they exceeded the levels established by the EU [11,25,48,57,58]. Concerning high Cd concentrations detected mostly in our NASSA specimens is mainly affected by the pollution of the cities in the North Aegean Sea coastline that became polluted due to marine, road and air traffic as well as due to domestic and industrial waste [30,32]. It may be concluded that *T. alalunga* may contain toxic metal Cd above the EU limits. More information is necessary for Cd levels in albacore from the Mediterranean Sea.

In the literature, lead bioaccumulation in tuna species shows highly variable results. Mol et al. [34] detected the content of 0.220 ± 0.007 mg kg^−1^ wet wt as highest Pb content in *T. alalunga* specimens caught from the Eastern Mediterranean, in contrast to the highest Pb content detected in the present study (0.557 mg kg^−1^ wet wt). Storelli [45] and Besada et al. [49] reported the highest Pb content in *T. alalunga* caught from the central Mediterranean Sea and the Atlantic Ocean as 3.130 and 0.012 mg kg^−1^ wet wt, respectively. Likewise, Pb contents were measured below the EU limits in tuna species caught from the Ionian Sea in the Mediterranean Sea and in the Atlantic Ocean [49,55,59]. Moreover, Ganjavi et al. [60] and Agah et al. [61] showed that the main sources of fish for Iranian tuna processing industry are the Persian Gulf and Oman Sea, where the possibility of Pb pollution is high due to heavy trafficking of oil and marine accidents. Agyekum Akwasi et al. [57] attributed the high Pb level in processed tuna from Ghana to the Pb load in the environment. In fact, many studies describe an accumulating trend for Pb in the edible part of tuna e.g., [11,25]. Nevertheless, albacore do not seem to be an important contamination source of Pb for humans.

### 4.2. Possible Human Health Risks

Out of 82 mercury THQs values calculated for Aegean Sea albacores, 80 were >1 until maximum 5.040 and therefore they were of concern (Table 1). Similarly as in the present study, Storelli [47] studied potential health risks from Hg via eating of seafood (different fish, cephalopod and crustacean species) and found that the health risk was mainly ascribed to consumption of albacore (THQ_Hg_ = 1.870) and some benthic fish, such as rosefish, *Helicolenus dactylopterus* (1.500) and thornback ray, *Raja clavata* (1.040). Also, the TEQs values caused by consuming frostfish, *Lepidopus caudatus* (0.710), horse mackerel, *Trachurus trachurus* (0.820) and striped mullet, *Mullus barbatus* (0.840) were rather high being close to 1 [47]. As shown in Table 1, cadmium and lead THQs were <1 in all cases suggesting the absence of health risks through consumption of the albacore. Mean cadmium THQ index from the present study (0.072) was higher compared to that reported in the previous study [45] for Mediterranean albacore (0.030). On the contrary, mean lead THQ index from our study was calculated at lower levels (0.006) to that reported in the literature (0.180) [47]. Moreover, except one albacore from the SASSA area, all the TTHQ index values from our study resulted in excess, indicating that the consumers could experience adverse health effects due to consumption of Aegean Sea albacore.

Although the THQs index calculation does not provide a quantitative estimate of the dangerous health effects of the exposed population, this methodology offers preliminary information on the potential health risk resulting from consumption of the albacore organisms. Specifically, risks associated with exposure to Hg, are mainly derived from consumption of fish.

## 5. Conclusions

The current study revealed that Hg had the highest mean concentration among tested elements, followed by Cd and Pb either in NASSA or SASSA Aegean Sea albacores. However, Hg concentrations of all samples were below the legal EU limit (2006). In contrast, TTHQs values due to consumption of Aegean Sea albacore indicated that human health risk might be of concern. Consumers should take this result into consideration. Regarding EU limits (2006), twenty-eight samples (34.2%) contained Cd and three samples (3.7%) contained Pb higher than the limit. According to the determined THQs for the elements Cd and Pb, Aegean Sea albacore tuna consumption can be considered safe for public health because this parameter did not exceed one unit, which is established as a potential health risk to the consumers. The Food and Agriculture Organisation (FAO) and the World Health Organisation (WHO) both recommend that governments monitor the levels of toxic metals in food. From the results of the present study, it might be concluded that the concentrations of trace metals, especially Hg, in albacore must be controlled comprehensively and periodically with respect to consumer health. Monitoring programs on metal levels in albacore must be conducted, and the consumer should be provided with such kind of information.

## Figures and Tables

**Figure 1 ijerph-16-00821-f001:**
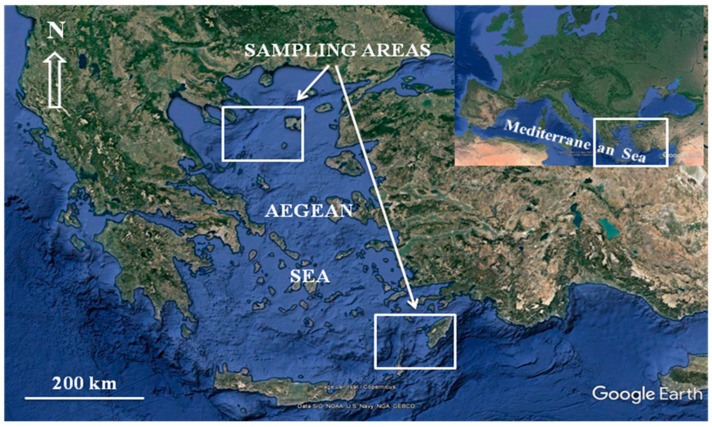
Sampling areas of the albacores (Northern and Southeastern Aegean Sea—Greece).

**Figure 2 ijerph-16-00821-f002:**
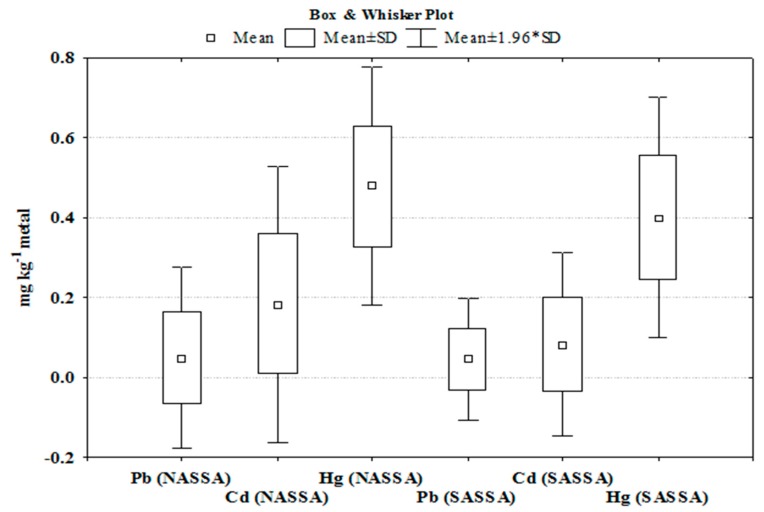
Graphical representation of box and whisker plots show data sets distribution of Hg, Cd and Pb levels in tested albacore tuna NASSA and SASSA samples. (*) corresponds to ± 1.96 times the SD.

**Figure 3 ijerph-16-00821-f003:**
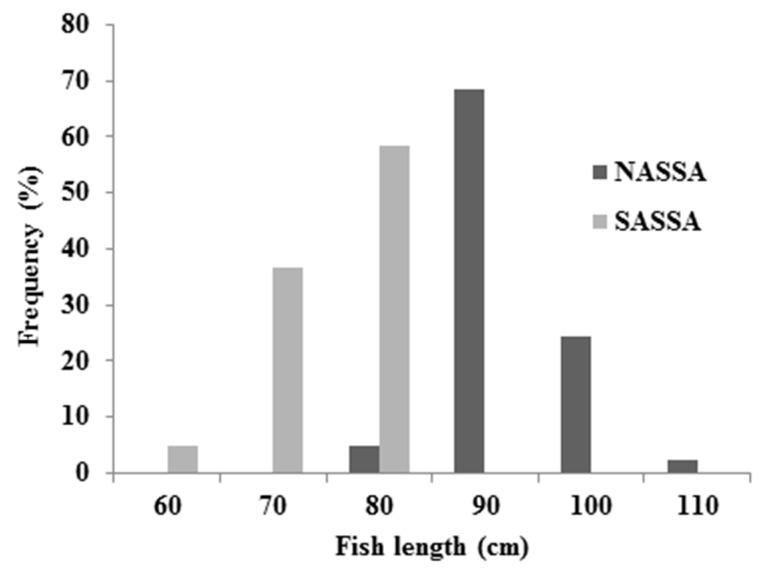
Length frequency distribution for NASSA and SASSA albacore samples.

**Figure 4 ijerph-16-00821-f004:**
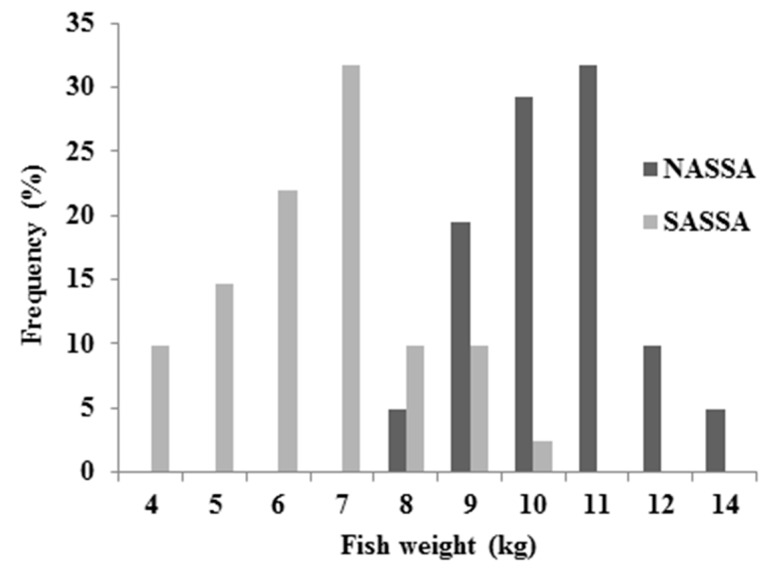
Weight frequency distribution for NASSA and SASSA albacore samples.

**Figure 5 ijerph-16-00821-f005:**
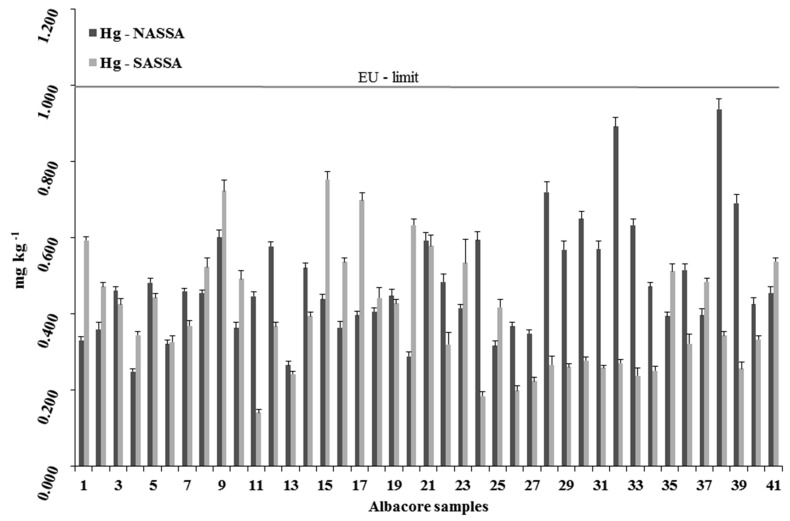
Mercury concentrations in each individual *T. alalunga*, caught from NASSA and SASSA areas.

**Figure 6 ijerph-16-00821-f006:**
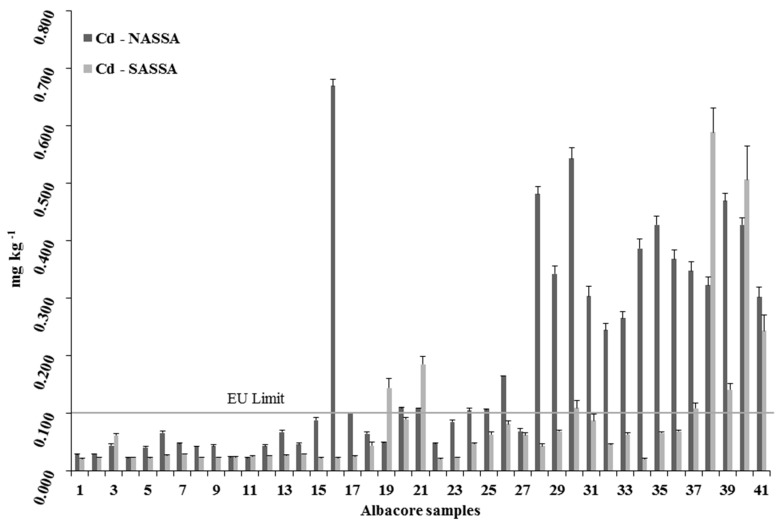
Cadmium concentrations in each individual *T. alalunga*, caught from NASSA and SASSA areas.

**Figure 7 ijerph-16-00821-f007:**
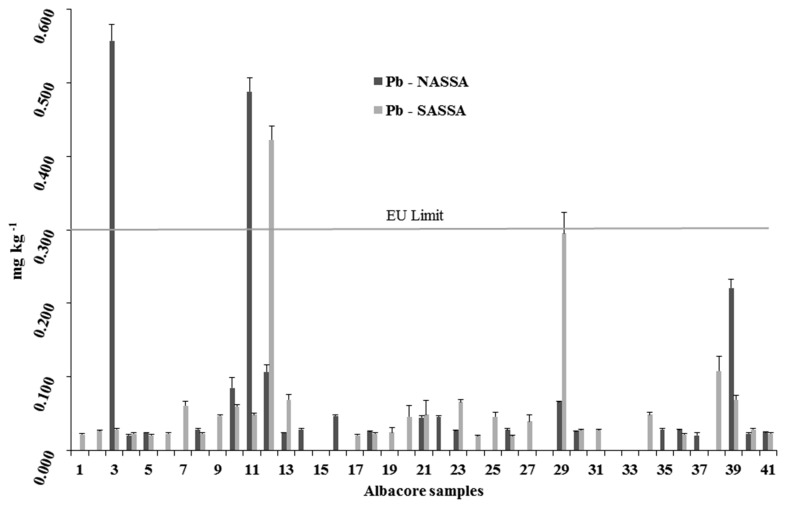
Lead concentrations in each individual *T. alalunga*, caught from NASSA and SASSA areas.

**Table 1 ijerph-16-00821-t001:** Values of target hazard quotient (THQ) and total target hazard quotient (TTHQ) per individual albacore (*T. alalunga*), caught from NASSA and SASSA areas, Greece (*).

Albacores	NASSA Area				SASSA Area			
	THQ_Hg_	THQ_Cd_	THQ_Pb_	TTHQ	THQ_Hg_	THQ_Cd_	THQ_Pb_	TTHQ
1	**1.779**	0.015	0.000	**1.794**	**3.186**	0.011	0.003	**3.201**
2	**1.934**	0.016	0.000	**1.950**	**2.536**	0.012	0.004	**2.552**
3	**2.482**	0.024	0.075	**2.581**	**2.289**	0.033	0.004	**2.326**
4	**1.338**	0.012	0.003	**1.353**	**1.854**	0.012	0.003	**1.869**
5	**2.595**	0.022	0.003	**2.620**	**2.386**	0.012	0.003	**2.400**
6	**1.725**	0.035	0.000	**1.760**	**1.757**	0.015	0.003	**1.775**
7	**2.472**	0.025	0.000	**2.497**	**1.983**	0.016	0.008	**2.006**
8	**2.450**	0.023	0.004	**2.477**	**2.816**	0.012	0.003	**2.831**
9	**3.229**	0.023	0.000	**3.252**	**3.890**	0.012	0.006	**3.909**
10	**1.956**	0.013	0.011	**1.980**	**2.644**	0.013	0.008	**2.665**
11	**2.402**	0.012	0.066	**2.479**	0.758	0.013	0.007	0.778
12	**3.106**	0.023	0.014	**3.143**	**1.977**	0.014	0.057	**2.048**
13	**1.429**	0.036	0.003	**1.468**	**1.300**	0.015	0.009	**1.324**
14	**2.799**	0.025	0.004	**2.828**	**2.122**	0.016	0.000	**2.138**
15	**2.359**	0.047	0.000	**2.406**	**4.046**	0.012	0.000	**4.058**
16	**1.961**	0.359	0.006	**2.327**	**2.885**	0.012	0.000	**2.897**
17	**2.133**	0.054	0.000	**2.187**	**3.756**	0.013	0.003	**3.772**
18	**2.187**	0.034	0.003	**2.225**	**2.380**	0.024	0.003	**2.407**
19	**2.407**	0.026	0.000	**2.433**	**2.300**	0.077	0.003	**2.380**
20	**1.553**	0.059	0.000	**1.611**	**3.401**	0.048	0.006	**3.455**
21	**3.192**	0.058	0.006	**3.256**	**3.111**	0.099	0.007	**3.217**
22	**2.606**	0.025	0.006	**2.637**	**1.719**	0.011	0.000	**1.731**
23	**2.235**	0.046	0.004	**2.285**	**2.875**	0.012	0.009	**2.896**
24	**3.197**	0.056	0.000	**3.253**	0.994	0.025	0.003	**1.022**
25	**1.703**	0.057	0.000	**1.760**	**2.246**	0.034	0.006	**2.286**
26	**1.977**	0.088	0.004	**2.069**	**1.069**	0.044	0.003	**1.115**
27	**1.875**	0.037	0.000	**1.912**	**1.204**	0.033	0.005	**1.242**
28	**3.869**	0.258	0.000	**4.127**	**1.435**	0.023	0.000	**1.458**
29	**3.057**	0.184	0.009	**3.250**	**1.402**	0.037	0.040	**1.479**
30	**3.493**	0.292	0.003	**3.788**	**1.494**	0.059	0.004	**1.556**
31	**3.068**	0.163	0.000	**3.231**	**1.392**	0.047	0.004	**1.442**
32	**4.793**	0.131	0.000	**4.924**	**1.456**	0.025	0.000	**1.481**
33	**3.407**	0.142	0.000	**3.549**	**1.279**	0.034	0.000	**1.313**
34	**2.541**	0.207	0.000	**2.749**	**1.349**	0.011	0.007	**1.367**
35	**2.122**	0.229	0.004	**2.356**	**2.751**	0.035	0.000	**2.787**
36	**2.773**	0.198	0.004	**2.974**	**1.730**	0.037	0.003	**1.770**
37	**2.133**	0.186	0.003	**2.322**	**2.606**	0.058	0.000	**2.664**
38	**5.040**	0.173	0.000	**5.213**	**1.848**	0.316	0.015	**2.179**
39	**3.713**	0.252	0.030	**3.995**	**1.381**	0.076	0.009	**1.466**
40	**2.289**	0.229	0.003	**2.521**	**1.795**	0.272	0.004	**2.070**
41	**2.450**	0.162	0.003	**2.616**	**2.891**	0.131	0.003	**3.024**
MIN	**1.338**	0.012	0.000	**1.353**	0.758	0.011	0.000	0.778
MAX	**5.040**	0.359	0.075	**5.213**	**4.046**	0.316	0.057	**4.058**
MEAN	**2.581**	0.099	0.007	**2.687**	**2.153**	0.044	0.006	**2.204**

(*) bold highlighted values exceed recommendations.

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
