# Peer review of "Bioaccumulation Levels and Potential Health Risks of Mercury, Cadmium, and Lead in Albacore (Thunnus alalunga, Bonnaterre, 1788) from The Aegean Sea, Greece"

_ijerph, 2019, doi:10.3390/ijerph16050821_

Round 1
Reviewer 1 Report
The paper investigated Hg, Cd and Pb concentrations in Albacore tuna from Aegean Sea. The measured metal concentrations were also compared to EU limit. Overall the presentation of the data is sound and the conclusion is reasonable. I suggest the author consider my comments below and the paper will be in a better shape for publication.
It will be more informative if the fish weights or the length can be incorporated in the data interpretation although the authors have showed two figures of weight and length distribution. As the heavy metal concentration tends to be higher in more grown-up fishes, that's why many adult fishes are not edible. It might be more appropriate if the metal concentrations can be normalized to weight or lengths, to indicate the growth factor. Otherwise it is not very clear if an high concentration indicates high exposure or just accumulation along the growth.
When Calculating THQ values, is there a reason why the exposure day is set to 365? To maximize the potential?
Please improve the figure quality. The fonts are vague.
Author Response
Response to Reviewer 1 Comments
Point 1: It will be more informative if the fish weights or the length can be incorporated in the data interpretation although the authors have showed two figures of weight and length distribution. As the heavy metal concentration tends to be higher in more grown-up fishes, that's why many adult fishes are not edible. It might be more appropriate if the metal concentrations can be normalized to weight or lengths, to indicate the growth factor. Otherwise it is not very clear if an high concentration indicates high exposure or just accumulation along the growth.
Response 1: This is a very interesting comment but we thing that data normalization has already achieved since the statistical method used is the Spearman Rank Order Correlations. According to this, the values were transformed into ranks and the strength and associations were measure between two raked variables. The significant correlations derived from this method for p < 0.05 indicate bioaccumulation, while on the other hand, non-correlated Pb in all areas and Cd in SASSA samples reveal that the concentrations of these elements were mainly associated to different parameters (i.e. pollution), rather than bioaccumulation.
To test this hypothesis data normalization was held as suggested by the reviewer. To examine whether a metal concentration associates mainly to bioaccumulation or other events, normalization of metal concentrations were held in terms of the total length of the fish for each individual. Normalizing the values in terms of total weight gives similar results since these two parameters are strongly related. The methodology used is the min-max normalization method slightly modified:
yi = (xi + xmin) / (xmax- xmin)
where yi is the derived normalized value, xi is the metal concentration for each individual. The modification of the method comes from the fact that the minimum (xmin) and maximum (xmax) concentrations used in the above equation were considered to be those from the smaller and larger individual, respectively, and not the statistical minimum and maximum concentration found from the sampling pool. This method can give negative, or even above unity values indicating whether bioaccumulation effect or not explains the metal concentrations in the individuals. Similar to the non-normalized values, a Spearman non-parametric rank test was performed to the normalized values in order to investigate the statistical differences between biological and chemical parameters in the examined areas.
After TL incorporation into the metal concentrations, the results of this normalization procedure showed that the derived Spearmen Ranked r values are identical with the r values presented in Tables S3 and S4. This occurs because the classified ranks of the normalization method are exactly the same with the ranks the variable inherit in the original methodology used in the manuscript.
Point 2: When Calculating THQ values, is there a reason why the exposure day is set to 365? To maximize the potential?
Response 2: Yes. The reason to set 365 days of exposure is to maximize the potential.
Point 3: Please improve the figure quality. The fonts are vague.
Response 3: The correction was made. Fonts on the new figures are set to a bigger size and the Tagged Image File Format (TIFF) was used for improving the figure quality.

Reviewer 2 Report
Review ijerph-451762
Title: Bioaccumulation Levels and Potential Health Risks of Mercury, Cadmium, and Lead in Albacore (Thunnus alalunga, Bonnaterre, 1788) from the Aegean Sea, Greece
Comments from reviewer
Overall, it is an interesting paper assessing the potential human health risks, concentrations of mercury (Hg), cadmium (Cd), and lead (Pb) via edible muscle samples from 82 individual albacores. However, I have some major questions about the paper that need to clarifications.
1. In the abstract, it says both cadmium and lead were above the permissible limits in 28 and 3 albacores respectively, whereas none of the samples contained mercury above the limit. However, next sentences said only mercury THQ and TTHQ values showed human health risk problems, while Cd and Pb THQ were not problems. Please clarify me on how to link and interpret these opposite results (Cd and Pb were over the permissible limits, but later only Hg became only heavy metal for human health risk). In other words, you’ll need to include sentences or paragraphs in the discussion section to show rationale of using concentration of each heavy metal in albacore and THQ- the results of each measure seem contradictory to one another.
2. My next question on this lead to formulas, US-EPA introduced in section 2.3. Even if 28 and 3 samples were over the permissible limits for Cd and Pb, but Cd and Pb were not detected any potential human health risks due to the consumption, can we possibly say this formula can be a reliable measure or link for the heavy metal assessment in albacore?
3. In the introduction line 46-47, it says “Tuna fish is very important……….” Only tuna fish is important in the human diet? Please revise the sentence to make it sounds more logical.
4. Section 3.3 Health risk assessment should be moved to the method section or could be deleted if it is already mentioned in the method section. Otherwise, you can merge 3.3 with 3.3.1 and name them together as Health risk assessment. Please revise.
5. As authors mentioned in the discussion, more information is necessary for Hg, Cd, and Pb levels in albacore from the Mediterranean Sea to identify reliable sources of origin of each heavy metal in albacore. For future study, PCA (Principal Component Analysis) could be utilized to see which areas have high levels of all three contaminants to better infer potential contamination sources.
Author Response
Response to Reviewer 2 Comments
Point 1: In the abstract, it says both cadmium and lead were above the permissible limits in 28 and 3 albacores respectively, whereas none of the samples contained mercury above the limit. However, next sentences said only mercury THQ and TTHQ values showed human health risk problems, while Cd and Pb THQ were not problems. Please clarify me on how to link and interpret these opposite results (Cd and Pb were over the permissible limits, but later only Hg became only heavy metal for human health risk). In other words, you’ll need to include sentences or paragraphs in the discussion section to show rationale of using concentration of each heavy metal in albacore and THQ- the results of each measure seem contradictory to one another.
Response 1: We agree with the suggestion of the reviewer and we changed in the revised abstract sentences in the lines between 21 and 24, as follows: “Potential health risks to human via dietary intake of albacore were estimated by the total target hazard quotients (TTHQs), which indicated that the consumers could acquire health problems due to consumption of Aegean Sea albacore”. Concerning the use of the present US-EPA formula in our paper, we have to add that this formula is acceptable from the most of the other researchers in the field (e.g. Chien et al., 2002; Copat et al., 2013; Yabanli et al., 2016; Varol et al., 2017; Núñez et al., 2018). Thus, our results for the THQs values are comparable to those available in the literature. Concerning the interpretation of the opposite results (Cd and Pb were above the permissible limits, but later only Hg became the only heavy metal harmful for human health), we have to notice the following: in the US EPA formula, only the reference oral dose (RfD) in mg kgbw-1 d-1 (which is: 1x10-4 for Hg, 1x10-3 for Cd, 4x10-3 for Pb) and the metals concentration in mg kg-1 are variables. For example, when we take into account the three max Pb, Cd and Hg concentrations in the NASSA samples (0.557, 0.669 and 0.938 ppm respectively), which are as numbers close to each other, the resulting THQs values are 0.075, 0.360 and 5.040 respectively (well differentiated numbers to each other).
Moreover, according to the reviewer’s suggestion and in addition to our revision in the abstract, we replaced the phrase discussing the THQs of mercury (Line 323) with other containing the TTHQs, in order to remove inconsistency between established EU limits and THQs values in our work.
Line 323. Phrase “In contrast, mercury THQs…..” was replaced with the phrase “In contrast, TTHQs…..”
Point 2: My next question on this lead to formulas, US-EPA introduced in section 2.3. Even if 28 and 3 samples were over the permissible limits for Cd and Pb, but Cd and Pb were not detected any potential human health risks due to the consumption, can we possibly say this formula can be a reliable measure or link for the heavy metal assessment in albacore?
Response 2: Of course is this US EPA formula a reliable measure for the heavy metal assessment in Aegean Sea albacores, given valuable results on TTHQs. US EPA and FDA warned that certain species of tuna, such as albacore, are specially known to accumulate Hg more than any other metal. This fact can possibly be the main reason that the US EPA formula is more “sensitive” to the Hg- as to Pb- and Cd- contents.
Point 3: In the introduction line 46-47, it says “Tuna fish is very important……….” Only tuna fish is important in the human diet? Please revise the sentence to make it sounds more logical.
Response 3: The correction was made. The sentence was changed to “Tuna fish, like other marine fish species, is very important......”
Point 4: Section 3.3 Health risk assessment should be moved to the method section or could be deleted if it is already mentioned in the method section. Otherwise, you can merge 3.3 with 3.3.1 and name them together as Health risk assessment. Please revise.
Response 4: The correction was made. Paragraph 3.3 was deleted and 3.3.1 was re-numbered to 3.3.
Point 5: As authors mentioned in the discussion, more information is necessary for Hg, Cd, and Pb levels in albacore from the Mediterranean Sea to identify reliable sources of origin of each heavy metal in albacore. For future study, PCA (Principal Component Analysis) could be utilized to see which areas have high levels of all three contaminants to better infer potential contamination sources.
Response 5: Thank You very much for this suggestion. Probably, a review paper with recently published data of the above three metals in albacore organisms from the whole Mediterranean Sea, coupled with PCA, could give useful information to the reader, about different potential contamination sources.

Round 2
Reviewer 2 Report
Review ijerph-451762_v2
Title: Bioaccumulation Levels and Potential Health Risks of Mercury, Cadmium, and Lead in Albacore (Thunnus alalunga, Bonnaterre, 1788) from the Aegean Sea, Greece
Comments from reviewer
Thanks for addressing my comments and the paper has been improved throughout. You have clarified my confusion about inconsistency between the permissible limits and THQ. You have mentioned that
“we have to notice the following: in the US EPA formula, only the reference oral dose (RfD) in mg kgbw-1 d-1 (which is: 1x10-4 for Hg, 1x10-3 for Cd, 4x10-3 for Pb) and the metals concentration in mg kg-1 are variables. For example, when we take into account the three max Pb, Cd and Hg concentrations in the NASSA samples (0.557, 0.669 and 0.938 ppm respectively), which are as numbers close to each other, the resulting THQs values are 0.075, 0.360 and 5.040 respectively.”
In other words, this fact can possibly be the main reason that the US EPA formula is more “sensitive” to the Hg- as to Pb- and Cd- contents. Overall, this revised version well addressed all my comments about the manuscript.